# Human capital's dual impact: Advancing innovation and technology diffusion in ASEAN-5 through the Nelson-Phelps-Romer Lens

Nguyen Ngoc Thach *

Department of International Economics, Ho Chi Minh University of Banking, Ho Chi Minh City, Vietnam

* thachnn@hub.edu.vn

## Abstract

The advanced ASEAN nations—Indonesia, Malaysia, the Philippines, Singapore, and Thailand—are navigating significant global uncertainties that challenge their industrialization ambitions. Human capital, recognized as a pivotal driver of technological progress, has not been adequately integrated into growth models for these countries. This study investigates the dual function of human capital within an extended Nelson-Phelps framework of technology diffusion, incorporating Romer's insights, across these five ASEAN countries from 1965 to 2019. Employing a Bayesian hierarchical analysis with specific informative priors effectively addresses statistical challenges. The findings reveal that human capital accelerates both domestic innovation and the adoption of foreign technologies in these nations. Notably, high-skilled labor significantly contributes to technological advancements, and domestic innovation plays a more substantial role in enhancing productivity growth than technology imitation. The extended Nelson-Phelps framework, which incorporates human capital's role in both innovation and technology diffusion, is well-suited for modeling the catch-up development of ASEAN economies. These insights offer valuable contributions to growth literature and practical applications in technology catch-up strategies.

## Introduction

Enormous productivity differentials between advanced and emerging or developing economies, alongside varying trends of convergence and divergence, have long characterized the global economy [1]. Persistent gaps in income, productivity, and technology, as well as their interconnections, remain critical concerns for policymakers and scholars. These disparities have spurred a vast body of literature focusing on the main long-term driver of the income gap: differences in technological capabilities [2]. Overviews of these trends, as highlighted by Durlauf et al. [3] and Acemoglu [4], underscore the period between 1960 and 2000 as one of widening disparities. While many countries have experienced economic growth, income inequalities have

**Data availability statement:** We confirm that all sources of data were presented in the article. We do not own data, and we do not have any special access privileges that others would not have. Real GDP, Mincerian human capital, TFP, physical capital stock, and employment are calculated from the Penn World Table 10.01, while school enrollment rates, government spending on education, exports as a percentage of GDP, and rule of law indicators are obtained from the World Development Indicators. These datasets are available at: https://www.rug.nl/ggdc/productivity/pwt/ and https://databank.worldbank.org/source/world-development-indicators.

**Funding:** The author(s) received no specific funding for this work.

**Competing interests:** The authors have no competing interests.

persisted, with distinct winners and losers emerging. High performers, such as East and Southeast Asian countries, have sustained exceptional growth rates, while regions like sub-Saharan Africa and parts of South and Central America have faced stagnation or volatility, marked by frequent output collapses. The ASEAN-5 nations—Indonesia, Malaysia, the Philippines, Singapore, and Thailand—have long aspired to transition into high-tech, competitive, and industrialized societies. However, global uncertainties have posed significant challenges to this ambition, leading to economic fluctuations, slowdowns, and periods of stagnation in productivity over the past decades (Fig 1). Policymakers must therefore base growth strategies on robust empirical models capable of addressing these vulnerabilities and charting a sustainable long-term trajectory.

The literature on technology catch-up, which gained prominence in the 1980s, offers insights into the role of technology in economic growth. Technology, being firm-specific yet systemic, is studied at macro, mezzo, and micro levels, as explored within the neo-Schumpeterian economics framework [5]. A critical question in this literature concerns the drivers of technology progress within a country and the diffusion of technology from leaders to followers. The Nelson-Phelps model of technology catch-up identifies education as the key factor in narrowing technological gaps among nations. However, earlier empirical studies estimating this model using frequentist econometric techniques have faced challenges, including multicollinearity, reverse causation, and heterogeneity [6–9]. These limitations have undermined the reliability of their findings.

Despite the rich body of research on ASEAN's economic development [10,11], among others), gaps remain in modeling the dynamics of technology diffusion. Specifically, no studies to date have applied a Nelson-Phelps framework of international

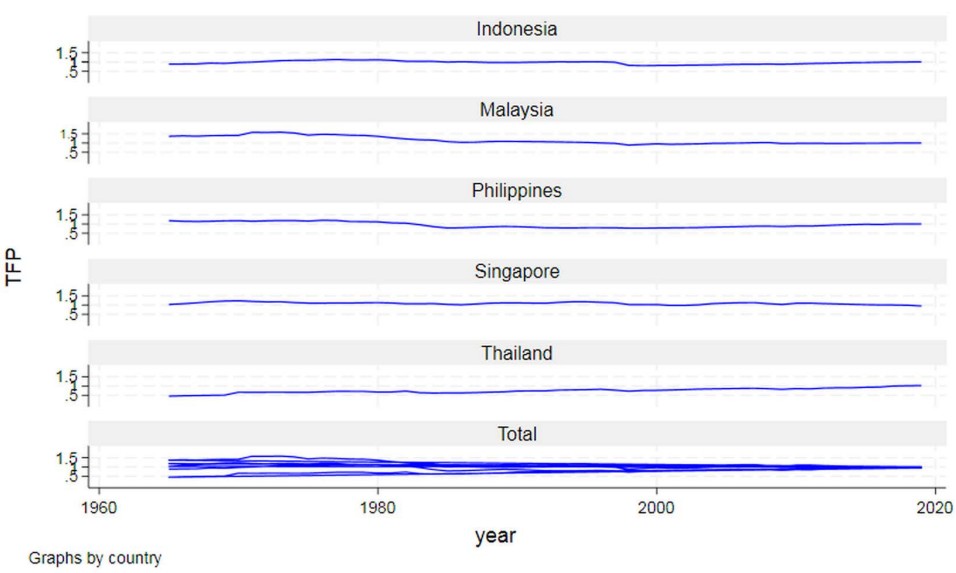

**Fig 1. Total factor productivity (TFP) trends in ASEAN-5.** *Source: Calculation by the author.*

technology diffusion to ASEAN economies. This gap leaves policymakers without actionable insights grounded in a robust growth model that accounts for the region's unique challenges and aspirations.

This study addresses this gap by exploring the dual role of human capital within an extended Nelson-Phelps technology diffusion framework through a Bayesian hierarchical approach. Unlike frequentist methods, the Bayesian framework efficiently handles statistical challenges, such as multicollinearity and heterogeneity, by leveraging shrinkage effects to produce robust and reliable results. The analysis uses a panel dataset of ASEAN-5 countries over the period 1965–2019, incorporating Romer's Schumpeterian growth model to enrich the Nelson-Phelps perspective.

The contributions of this research are fivefold. First, the adoption of a Bayesian hierarchical method provides a flexible and effective approach to addressing statistical complexities, yielding robust and reliable empirical insights. Second, the findings emphasize the dual role of human capital in fostering innovation and facilitating the adoption, implementation, and imitation of foreign technologies, consistent with the combined Nelson-Phelps and Schumpeterian models. Third, the study underscores the greater contribution of skilled labor (secondary and tertiary education) to innovation and technology transfers compared to unskilled labor. Fourth, it highlights that national innovations play a more significant role in boosting productivity than technology adoption in the ASEAN-5 economies. Fifth, the extended Nelson-Phelps model, enhanced with Romer's insights, is demonstrated to be a suitable framework for promoting sustainable industrialization and technological advancement in ASEAN countries. These contributions provide valuable insights for policymakers in formulating effective growth strategies.

The remainder of the paper is structured as follows. Section II reviews Romer's Schumpeterian growth model, the Nelson-Phelps framework of international technology diffusion, and related empirical studies. Section III details the Bayesian hierarchical regression methodology, model specifications, and data sampling. Section IV presents and analyzes the empirical findings from both traditional growth accounting and the extended Nelson-Phelps model. Finally, Section V concludes the study by summarizing its findings and implications.

## Literature review

### Analytical framework

The global economy exhibits pronounced productivity disparities between advanced and emerging or developing economies, accompanied by divergent convergence patterns [1]. Persistent gaps in income, productivity, and technological capabilities, along with their interconnectedness, have remained a central focus for policymakers and scholars. These disparities have spurred extensive research into the primary long-term determinant of income inequality—variations in technological capabilities [2]. There is a growing literature on technology diffusion since the 1980s. During this decade, this academic area was divided into two almost unrelated strands: macroeconomic growth and technological capabilities. Over the 1990s, while the technology diffusion field began to surge immensely, these two strands, became close to each other, evolving further into the neo-Schumpeterian economics on technology progress and economic growth, which unlike the traditional economics, operates at all (macro-, mezzo-, and micro-) levels. Macroeconomic interdependence between technology and growth considerably enhances comprehensive strategies and policies of technology adoption and innovation advancements. The endogenous growth models [12,13] have emerged as the most influential ones within this area.

Romer's Schumpeterian growth model envisions an economy comprising three sectors: the final-good sector, the intermediate capital-good sector, and the R&D sector [13]. In the final-good sector, human capital (skilled labor) $H_Y$, unskilled (raw) labor $L$, and a variety of capital goods $x_j$ ($j = 1, 2, 3 \ldots$) are utilized to produce the final output $Y$. The production technology of this sector is represented by:

$$Y_t = H_{Yt}^{\alpha} L_t^{\beta} A_t x_t^{1-\alpha-\beta}.$$

(1)

where $\alpha$ and $\beta$ is the income share of human capital and unskilled labor, respectively.

The intermediate capital-good sector manufactures intermediate capital goods by utilizing the portion of output saved from the final-good sector, along with the knowledge generated by the R&D sector. Unlike the competitive nature of the final-good sector, firms in the intermediate capital-good sector operate as monopolists, holding patents for the knowledge they have either invented or acquired from the R&D sector. The quantity of capital utilized in year $t$ is given by:

$$K_t = nA_tx_t \tag{2}$$

$$\text{or } x_t = \frac{K_t}{nA_t} \tag{3}$$

where $n$ is units of $Y$ to produce one unit of any intermediate capital-good.

By substituting (3) into (1), we obtain:

$$Y_t = (A_tH_{Yt})^\alpha (A_tL_t)^\beta K_t^{1-\alpha-\beta} n^{\alpha+\beta-1}. \tag{4}$$

The R&D sector using human capital $H_A$ and the knowledge stock $A$ yields knowledge for the intermediate capital-good sector. The growth rate $\dot{A}$ of $A$ is:

$$\dot{A} = \phi H_{At} \tag{5}$$

where $\phi$ is a parameter.

We know:

$$H_{Yt} = \frac{1}{\phi} \frac{\alpha}{(\alpha+\beta)(1-\alpha-\beta)} i \tag{6}$$

where $i$ is fixed interest rate.

Assuming a fixed $H$, we have:

$$H_{At} = H - H_{Yt} = H - \frac{1}{\phi} \frac{\alpha}{(\alpha+\beta)(1-\alpha-\beta)} i = const. \tag{7}$$

Hence

$$\dot{A} = \phi H_{At} = \phi H - \frac{\alpha}{(\alpha+\beta)(1-\alpha-\beta)} i = g. \tag{8}$$

Along the steady state balanced growth path of the economy:

$$\dot{K} = \dot{Y} = \dot{C} = \dot{A} = g = \phi H - \frac{\alpha}{(\alpha+\beta)(1-\alpha-\beta)} i. \tag{9}$$

From the above, Romer's Schumpeterian model concludes that sustaining perpetual growth requires an economy to continuously expand its stock of knowledge. Achieving this primarily hinges on two key conditions: (1) increasing the ratio of human capital to unskilled labor $(\frac{H}{L})$ and (2) raising the proportion of human capital allocated to research relative to the total human capital $(\frac{H_A}{H_Y})$.

In their seminal article, Nelson & Phelps [12] offered growth economists with a novel analytic framework to delve into the role of human capital in speeding the process of technology diffusion and growth. It emphasizes the interaction

between human capital and the process closing the gap between a country's current technological level and the global technological frontier:

$$\frac{\dot{A}_i(t)}{A_i(t)} = \varnothing H_i(t)\left(\frac{A_m(t) - A_i(t)}{A_i(t)}\right)$$

(10)

From (10):

$$g_{A,i}(t) = \frac{\dot{A}_i(t)}{A_i(t)} = \varnothing H_i(t)\left(\frac{A_m(t)}{A_i(t)} - 1\right)$$

(11)

where $A_i(t)$ is the current technological level of the economy (e.g., TFP or labor productivity), $g_{A,i}(t) = \frac{\dot{A}_i(t)}{A_i(t)}$ is the rate of technological progress or the growth rate of technology, $H_i(t)$ is the stock of human capital in the economy, $A_m(t)$ is the global technological frontier (maximum attainable technological level at time $t$) or a leader country's technological level, $A_m(t) - A_i(t)$ is the technological gap (distance to the frontier), and $\varnothing$ is a parameter capturing the efficiency of technology adoption.

(10) and (11) show that the growth rate of technology ($g_{A,i}(t)$) is proportional to the relative technological gap and the level of human capital ($H_i(t)$). The larger the gap between the global technological frontier ($A_m(t)$)and the current level of technology ($A_i(t)$), the greater the potential for catching up. As the economy catches up with the technological frontier ($A_i(t) \to A_m(t)$), the growth rate of technology ($\frac{\dot{A}_i(t)}{A_i(t)}$) declines. Human capital enhances the economy's ability to absorb and implement advanced technology. The greater the stock of $H_i(t)$, the faster the country can close the technological gap. Higher levels of human capital ($H_i(t)$) increase the rate of convergence to the frontier, accelerating technological progress. Shortly, countries with low initial technology ($A_i(t)$) but high human capital ($H_i(t)$) can grow faster by adopting frontier technologies and catch up more quickly.

The model [6] builds on the models [12] and [13] by incorporating human capital into both technological adoption and endogenous innovation. In their framework, human capital plays a dual role: facilitating the adoption of technology from the global frontier and contributing to the creation of new technologies within an economy. Benhabib and Spiegel's model assumes that the growth of a country's productivity, ($A_i(t)$), is driven by two key mechanisms: first, the adoption of foreign technology, which reflects the Nelson–Phelps idea of closing the technological gap; and second, domestic innovation, which incorporates Schumpeterian insights into the direct role of human capital in generating new technologies [6]. The problem is mathematically formulated as follows:

$$\frac{\dot{A}_i(t)}{A_i(t)} = \varnothing_1 H_i(t) + \varnothing_2 H_i(t)\left(\frac{A_m(t) - A_i(t)}{A_i(t)}\right)$$

(12)

where $\varnothing_1$ is a parameter capturing the role of human capital in domestic innovation, $\varnothing_2$is a parameter capturing the effectiveness of human capital in technology adoption.

In (12), the term $\varnothing_1 H_i(t)$ reflects the contribution of human capital to innovation and the development of new technologies. Higher levels of human capital enhance the ability to innovate and sustain growth domestically, even when the country has already caught up to the technological frontier. While $\varnothing_2 H_i(t)\left(\frac{A_m(t) - A_i(t)}{A_i(t)}\right)$ represents the contribution of human capital to closing the gap between the global technological frontier and the country's current technology level. Shortly, countries with higher levels of human capital experience faster productivity growth through both technology adoption and innovation. Countries farther from the technological frontier (large $[A_m(t) - A_i(t)]$) can grow faster if they have sufficient human capital to adopt advanced technologies. Conversely, for countries closer to the technological frontier, growth

depends more on domestic innovation, which requires significant human capital. As the gap between $A_m(t)$ and $A_i(t)$ narrows, the catch-up effect diminishes, and the growth rate becomes increasingly reliant on innovation.

A conceptual diagram of the Nelson-Phelps model extended with Schumpeterian innovation concepts is presented in Fig 2.

## Related empirical studies

The empirical literature on the linkage between human capital and technology diffusion has evolved into multiple distinct streams.

A conventional approach involves modeling human capital as an independent variable, without accounting for its interaction with technology variables, within aggregate production functions. This ranges from straightforward log-linear regressions to more complex mathematical formulations, see [14] and numerous subsequent replication studies. The influential work by Mankiw et al. [14] revived interest in the Solow growth model, emphasizing the need for accurate specification and rigorous assessment. Their analysis explained over half the cross-country income variance, except for OECD economies, where performance declined considerably. For non-oil (98) and intermediate (75) countries, the R-squared was 0.59, but for 22 OECD nations, it dropped to 0.01, with an insignificantly negative coefficient for ln(n+0.05) and an inflated capital elasticity (alpha=0.6). Critics [15,16] highlighted their two key issues: treatment of country-specific factors as error terms, risking omitted variable bias, and uniform assumptions for capital depreciation (0.03) and technology growth (0.02), both based on U.S. data. These limitations raised doubts about the robustness of their findings. Islam [16] criticized the heterogeneity issues inherent in Barro-type regressions, including those of [14], and advocated for a panel data framework. He highlighted that this approach better accounts for variations in aggregate production functions across different economies, yielding markedly different results compared to single cross-country regressions. Notably, his analysis revealed a capital share in income that was lower than anticipated. Temple [17], employing robust estimation methods to test the model of [14] across different country groups, argued this model should explain economic growth in developing, NICs, and OECD nations. However, excluding Portugal and Turkey from the OECD sample reduced the R-squared from 0.35 to 0.02, revealing weak explanatory power for the OECD's most cohesive group. When dividing the sample into quartiles, regressions showed acceptable fits (R-squared: 0.58–0.67) but exhibited significant parameter variation. Moreover, Young [18,19] demonstrated that the remarkable economic growth observed in East Asia was primarily driven by increases in labor and capital inputs rather than improvements in TFP. Similarly, Barro and Sala-i-Martin [20] provided evidence that the augmented Solow model aligns with the convergence rates they estimate both across countries and within regions in the U.S., Japan, and several European nations. This strand of growth literature, referred to by Young as

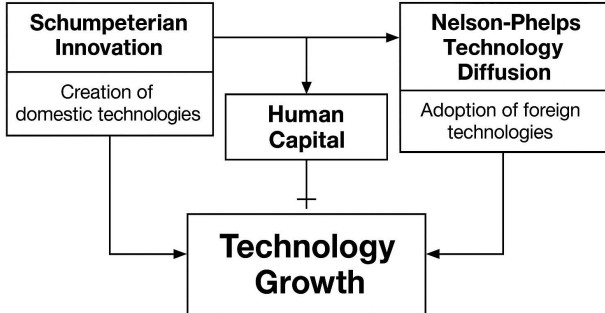

**Fig 2. The combined catch-up framework.** *Source: Drawn by the author.*

the "neoclassical revival," posits that productivity levels and growth rates are relatively uniform across countries, attributing disparities in output levels and growth rates primarily to variations in physical and human capital.

The relationship between human capital and technology creation, implementation, and adoption has been extensively explored within an increasingly sophisticated body of growth literature. Early contributions, including [21]–[23], highlighted the critical role of human capital in driving technological progress. Bartel and Lichtenberg [22], estimating labor demand functions, demonstrated that highly educated workers exhibit a comparative advantage in adapting to and implementing new technologies. Foster and Rosenzweig [23], using household-level panel data from rural India, showed that learning by doing and spillovers significantly enhance the adoption and profitability of high-yield seed varieties during the Green Revolution. Eaton and Kortum [24,25] emphasized the pivotal role of international technology diffusion in productivity growth. Using a cross-section of 19 OECD countries, Eaton and Kortum [24] found that, aside from the U.S., most OECD nations derived nearly all productivity growth from foreign sources. Extending their analysis to five advanced economies (France, Germany, Japan, the U.K., and the U.S.), Eaton and Kortum [25] identified the U.S. and Japan as driving at least two-thirds of productivity growth, proposing trade as a critical channel for embodied technological progress. They argued that eliminating international barriers to diffusion would not only converge productivity levels globally but also significantly boost productivity everywhere. Subsequent studies further deepened this understanding by integrating human capital into models of technology diffusion and economic growth. Klenow and Rodriguez-Clare [26], incorporating diverse measures of human capital (Mincerian, primary, secondary, tertiary schooling) into growth models, concluded that differences in productivity growth overwhelmingly explain disparities in growth rates. Hall and Jones [27], utilizing Mincer regression evidence to construct human capital stocks, corroborated the significance of productivity differences in explaining international income variation. Duffy and Papageorgiou [28], employing a constant-elasticity-of-substitution (CES) production function across 82 countries, revealed that physical and human capital-adjusted labor are more substitutable in wealthy nations than in poorer ones, where labor and capital tend to be complementary. Hanushek and Kimko [29] underscored the importance of labor-force quality, demonstrating that international test scores in mathematics and science are robustly linked to growth, even when excluding East Asian countries. Krueger and Lindahl [30] supported the positive effects of educational attainment on income, finding that after accounting for measurement error, the impact of education on income growth in cross-country data aligns with microeconometric estimates of the returns to schooling. More recent contributions have refined the theoretical foundations of these relationships. Hanushek et al. [31] developed state-level measures of worker skills in the U.S., incorporating cognitive achievements. Their findings revealed that differences in knowledge capital account for 20–30% of state-level variation in per capita GDP, with contributions evenly split between school attainment and cognitive skills. Collectively, this body of research underscores the intricate interplay between human capital, technology diffusion, and economic growth, offering critical insights into the mechanisms underpinning long-term productivity improvements.

A prominent strand of empirical research, initially developed by Nelson and Phelps [12] and subsequently expanded by Benhabib and Spiegel [6,32] with insights from [13], has increasingly concentrated on the dual role of human capital. This research investigates whether human capital directly impacts productivity by enhancing a nation's ability to innovate technologies tailored to domestic production while also influencing the rate of technological diffusion and the pace of catch-up with more advanced economies. Initially, Nelson and Phelps [12] proposed a catch-up model of international technology diffusion to explain economic growth. Their framework comprised two key components: (1) the growth of the technology frontier reflects the rate of innovation, while TFP growth depends on the adoption of these innovations, positively correlated with the gap between the technology frontier and current productivity. This formalized Gerschenkron's catch-up hypothesis [33]. (2) The rate at which this productivity gap closes is determined by the level of human capital. Crucially, Romer [13] postulated that human capital may directly influence productivity by determining the capacity of nations to innovate new technologies suited to domestic production. Relying on these models, [6], applying cross-country data to the extended Nelson–Phelps hypothesis, concluded that technological knowledge diffuses from leader economies

to their followers, with the speed of this diffusion being critically reliant on a nation's education levels. The drawback of the methodology employed in their research is its low capability of handling statistical complexities such as multicollinearity, heterogeneity and reverse causation, inherent in the neoclassical growth models, leading to estimation biases. Over a decade later, Benhabib and Spiegel [32] tested two specifications of technology diffusion based on the Nelson-Phelps hypothesis using cross-sectional data for 27 nations spanning 1960–1995. Their findings supported the logistic specification over the more restrictive exponential model. Revisiting the Nelson-Phelps hypothesis, Correa-López [34] introduced a growth model with quality-improving innovations, showing that increases in the stock of human capital, interpreted as the skill composition of non-research labor, raise steady-state growth rates and amplify sensitivity to interest rate variations when adoption lags are longer. The role of human capital in enhancing productivity through fostering innovation and technology adoption has also been substantiated in other studies [35,36]. Taking a different perspective, Vandenbussche et al. [37] argued that the effects of human capital on innovation and imitation are not uniform. They posited that different types of human capital—skilled workers (measured by secondary and tertiary education) and unskilled workers (measured by primary education)—influence these processes differently, depending on a country's relative distance to the technology frontier. This conclusion is supported by subsequent studies, including [38,39]. Cheng et al. [39], analyzing panel data for 16 Asian developing countries from 1970 to 2009, revealed that the aggregate level and composition of human capital significantly affect productivity growth via technology diffusion. Their findings exhibited that primary and secondary education positively influence TFP growth, while tertiary education exhibits a positive but statistically insignificant effect. This outcome is likely attributable to the relatively low average years of tertiary education in these countries, which limits its contribution to TFP growth. Furthermore, the interaction between primary education and the technology gap yielded a negative but insignificant coefficient, suggesting that low levels of education are insufficient for effectively absorbing foreign technology [40]. Conversely, secondary and tertiary education play crucial roles in closing the TFP gap for Asian developing countries, with both interaction terms being statistically significant. As countries approach the technology frontier, higher levels of education (secondary and tertiary) serve as essential absorptive capacities for adopting foreign technologies and narrowing the gap. In a somewhat different vein, AtiqurRahman and Zaman [41] revisited the human capital-based technological catch-up hypothesis proposed by Benhabib and Spiegel [6] for developing countries, utilizing panel data from 75 nations. Their analysis failed to identify any positive role for technological catch-up in driving economic growth when the U.S. was used as the global technological leader or when individual countries were designated as regional technological leaders. However, they found strong evidence supporting technological catch-up when OECD countries collectively, or major trading partners with higher incomes, were considered as technological leaders for developing nations. Akhvlediani and Cieślik [42] investigated the role of human capital in driving TFP growth across EU countries from 1950 to 2014, using a modified Nelson–Phelps framework. Their findings highlight the positive impact of human capital on technological progress and diffusion. However, the evidence does not support full convergence to the technological frontier, as peripheral EU countries exhibit weak convergence rates, largely due to inadequate investment in human capital. Cervellati et al. [43] offered new evidence that human capital plays a pivotal role in the cross-country diffusion of technologies. Higher levels of human capital are linked to shorter adoption lags and more intensive use of new technologies, thereby establishing a connection between educational attainment and the degree of economic backwardness.

In the Nelson–Phelps framework, the diffusion of disembodied technical knowledge from technology leaders to followers enhances TFP in the latter. However, the framework does not explicitly account for patent protection or blueprint ownership, necessitating alternative mechanisms to sustain innovation and mitigate free-riding behavior. Several theoretical models have directly examined the role of imitation costs, which dissipate rents from innovative activities. Grossman and Helpman [44], along with [45,46], explored a North-South framework where the North innovates under patent protection, while the South imitates at a cost due to lower labor expenses. Building on this foundation, Aghion et al. [47] proposed a leapfrogging model in which firms incur costs to catch up and surpass competitors, thereby capturing a larger share of profits. Eaton and Kortum [24,25] developed a model incorporating patenting costs, where patents reduce—though do

not eliminate—the risk of imitation. Barro and Sala-i-Martin [20,48] introduced a model of technological diffusion in which innovation costs are relatively low in leading economies, while imitation costs dominate in follower nations. Basu and Weil [49] suggested that technological barriers to imitation in follower countries, particularly those arising from significant differences in factor proportions between regions (e.g., North and South), could lead to the formation of "convergence clubs." These disparities may render imitation opportunities less "appropriate," failing to direct technological progress toward efficient cost-saving innovations [50]. Nonetheless, technology diffusion can still occur across convergence clubs, with imitation costs replacing patent protection as the mechanism sustaining innovation within these groups. Eeckhout and Jovanovic [51] proposed a model where imitators can adopt technology only after a time lag, creating implicit imitation costs that incentivize innovators to maintain their competitive edge. These models collectively underscore the necessity of both imitation costs and innovation incentives for technology diffusion to drive economic growth effectively. Thus, for the Nelson–Phelps framework to remain viable, it must assume an appropriate market structure that establishes an economic equilibrium, ensuring the coexistence of innovative activity and technology diffusion. More recently, Doré and Teixeira [8] examined Brazil's real convergence from 1822 to 2019 across three scenarios: (i) relative to six other Latin American countries (LA6), (ii) to Portugal, and (iii) to the technological frontier, the United States. Results from extended unit root tests support Brazil's very long-run real convergence with LA6 and Portugal, but not with the US. Additionally, the estimated Markov-switching models highlight the positive role of institutional quality in fostering Brazil's real convergence.

The empirical literature on technology diffusion has expanded substantially, despite persistent challenges in measurement. Griliches [52] underscored the presence of significant R&D spillovers, providing foundational evidence for the transmission of innovation. Coe and Helpman [53] demonstrated that foreign R&D investments positively influence domestic productivity, likely facilitated by the transfer of technological knowledge through international trade linkages. At a micro level, Branstetter [54], utilizing disaggregated data, identifies notable research spillovers among firms closely positioned within "technology space." Similarly, [55] highlight that the significance of R&D spillovers varies across countries; while domestic research predominantly explains productivity growth in the U.S., foreign R&D exerts a more substantial influence in nations like Italy and Canada. Xi Lin [56] showed that centrality in global value chains amplifies R&D spillovers—underscoring the necessity of human capital and network position in achieving convergence. Ito [9] used spatial convergence models in China to show that innovative human capital supports green-growth convergence across provinces, with thresholds echoing the Nelson–Phelps mechanism.

Economic growth plays a crucial role in advancing a developing nation's economy by creating well-paying jobs, enhancing the overall quality of life, and delivering numerous other benefits. When effectively managed, economic growth fosters the retention and expansion of employment opportunities, investments within communities, and catch-up industrialization. As a result, economic growth remains a top priority for governments and policymakers worldwide, and the ASEAN community, where most member states are follower economies, is no exception. There is broad empirical literature on development in ASEAN, see [11,57–63]. Despite the extensive body of research on the economic growth and development of ASEAN countries, the challenges of modeling catch-up growth remain largely unexplored, especially for the more advanced nations such as Indonesia, Malaysia, the Philippines, Singapore, and Thailand. To the best of our knowledge, no study has yet tested the Nelson-Phelps hypothesis of technology catch-up in the ASEAN context.

Based on the comprehensive survey above, particularly regarding analyses of the role of human capital in technology diffusion, several gaps are identified:

First, from a methodological perspective, earlier studies that incorporated human capital either as a direct input in production functions or indirectly through its influence on international technology diffusion relied on various frequentist techniques. These approaches struggle to address severe multicollinearity, which is inherent in conventional econometric models inspired by the Nelson-Phelps hypothesis. As highlighted in Subsection 4.1, initial income—a key control variable—exhibits high correlations with the three human capital measures (Mincerian human capital index, and school enrollment rates at the secondary and tertiary levels), with respective correlation coefficients of 0.9, 0.96, and 0.8. Additionally,

initial income is strongly correlated with other control variables. Reverse causation further complicates these analyses, as the dependent variable (the level and growth of per capita income) can influence human capital accumulation, as noted by Benhabib and Spiegel [6,32]. Moreover, excluding a uniform TFP level or growth, or omitting these variables from the Nelson-Phelps model—as done in [6]—introduces omitted variable bias.

Second, due to data limitations, cross-sectional (Barro-type) regressions are often used instead of panel regressions, which exacerbates heterogeneity issues in previous growth studies, see [6,14,32,64–66]. Heterogeneity undermines the stability of "Barro regressions," leading to biased results [67–71].

Third, despite significant efforts to promote catch-up industrialization in ASEAN countries over the decades, no growth studies have empirically tested the central hypothesis of Nelson and Phelps [12] within the ASEAN context. Existing analyses of economic growth in ASEAN primarily rely on simplified linear or log-linear regressions to examine the impact of selected variables on total output levels or growth rates. However, given the fluctuations in GDP observed over the past three decades in the ASEAN region [72], the findings of these studies and their policy recommendations have proven insufficient for fostering sustainable long-term growth. A more suitable growth model is urgently needed.

## Method, model, and data

**Method and model.** Since the 1990s, the Bayesian paradigm has immensely transformed numerous applied and fundamental disciplines, including the social, behavioral, and developmental sciences [73]. This shift has occurred against a backdrop of growing criticism of the frequentist paradigm, particularly concerning the reliance on p-values for statistical inference. A p-value represents the probability of obtaining data as extreme (or more extreme) as observed, assuming the null hypothesis is true [74]. However, the common practice of using a threshold of $p < 0.05$ for statistical significance has been criticized as arbitrary and potentially misleading. A large p-value does not necessarily indicate that the null hypothesis is likely true; rather, it suggests insufficient evidence to reject it [75,76]. Despite its widespread adoption in various fields, the Bayesian framework remains underexplored within growth research [77]. This study represents the first attempt to introduce Bayesian estimation to this domain, addressing challenges related to model uncertainty arising from intricate statistical complexities. Unlike traditional frequentist methods, Bayesian analysis provides a distinct approach to hypothesis testing [78–81]. One of its most notable advantages lies in Bayesian inferences generated through the Markov Chain Monte Carlo (MCMC) algorithm, which integrates prior knowledge with observed data to produce more reliable and robust parameter estimates. This approach not only mitigates issues of uncertainty but also yields the entire posterior distribution of the parameters, offering a comprehensive understanding of their behavior [82–84]. In employing a MCMC algorithm, the data D, which is a realization of $X \sim f(D|H)$, is integrated with prior information encapsulated in a prior distribution with density $\pi(H)$. This integration results in the posterior distribution $(H|D)$, summarizing the updated beliefs. The posterior distribution is derived from the joint distribution $(D|H) \pi(H)$ using Bayes' formula:

$$\pi(H|D) = \frac{f(D|H)\,\pi(H)}{\int f(D|H)\pi(H)dH}$$

(13)

where $m(D) = \int f(D|H)\pi(H)dH$ is the marginal density of $X$.

The primary advantage of the Bayesian approach, particularly Bayesian hierarchical methods, in growth econometrics lies in their ability to address the challenges posed by model uncertainty, often stemming from data quality issues such as multicollinearity, heterogeneity, and reverse causality. These challenges can compromise the stability and reliability of parameter estimates, undermining the validity of frequentist analyses. Bayesian hierarchical methods effectively mitigate multicollinearity by isolating the individual effects of covariates, offering a clear advantage over frequentist estimators [85–87]. In frequentist regressions, exact collinearity among regressors renders individual coefficients unidentified [87], leading to unstable estimates and inflated standard errors [88]. Even slight deviations from collinearity assumptions can

significantly distort frequentist estimates [86]. In contrast, Bayesian posterior means remain well-defined when informative priors are incorporated [87]. Furthermore, in cases of reverse causation, the Bayesian approach provides a robust solution by integrating prior information with observed data, thereby improving the estimation of effect sizes, relationships, and directions among variables. This synthesis results in a more precise and reliable parameter estimation process [85,86]. Notably, Bayesian hierarchical methods are superior because, while providing mean values, they simultaneously account for unity-specific effects, thereby yielding more precise parameter estimates [89].

This study employs Bayesian hierarchical regressions to examine the central hypothesis of technology catch-up within the Nelson-Phelps framework, as extended by Benhabib and Spiegel [6] and building on Schumpeterian foundational ideas on the role of skilled labor in fostering and implementing domestic innovations [13]. Addressing the connection between education and economic growth, Nelson and Phelps [12] argued that the conventional approach of directly including educational attainment as an input in the production function may fundamentally misrepresent the dynamic interplay between education and productivity growth. As highlighted by Benhabib and Spiegel [32], distinguishing between education's role as a factor of production and its function in facilitating technology diffusion carries critical policy implications. In the former, the economic benefit of education is confined to its marginal productivity, whereas in the latter, education influences the growth rate of TFP and technology diffusion, yielding cumulative benefits over future output levels. Hence, to rigorously identify the role of human capital in the growth process, we test two models: the traditional growth accounting framework and the extended Nelson-Phelps specification.

The traditional growth accounting model is estimated using a Bayesian hierarchical framework:

$$dq_{it} = a_0 + a_1 dX_{it} + a_2 Y_{it} + a_3 dZ_{it} + a_4 Q_o + u_i + \varepsilon_{it}. \tag{14}$$

Following the approach of Benhabib and Spiegel (1994), a Bayesian hierarchical model is specified for the extended Nelson-Phelps framework:

$$dq_{it} = b_0 + b_1 dX_{it} + b_2 Y_{it} + b_3 logZ_{it} + b_4 Q_o + v_i + \in_{it}. \tag{15}$$

where $dq$ denotes the annual growth rate of the dependent variable, per capita income. The core independent variables, drawn from the traditional growth accounting framework and the extended Nelson-Phelps model, include $dZ$, which represents the annual growth of human capital variables; $logZ$, the logarithm of mean human capital indices, and $Q_o$, the initial per capita income in 1964. Control variables are divided into two groups: $dX$, denoting the annual growth rates of TFP, physical capital stock ($K$), labor ($L$); and $Y$, representing trade openness ($Ex$) and institutional quality ($Ru$). The terms $u$ and $v$ are random effects capturing country-specific factors; $\varepsilon$ and $\in$ represent residual error terms. Indices $i$ and $t$ refer to countries (Indonesia, Malaysia, the Philippines, Singapore, and Thailand) and years (1965–2019), respectively.

(14) represents a single-equation specification that combines both level-1 and level-2 effects. To reformulate it in hierarchical terms, distinct equations are defined for each level:

$$dq_{it} = a_{0i} + a_1 dX_{it} + a_2 Y_{it} + a_3 dZ_{it} + a_4 Q_o + \varepsilon_{it}. \tag{16}$$

$$a_{0i} = a_{00} + u_{0i}. \tag{17}$$

Here, the intercept equation $a_{0i}$ is composed of the overall mean intercept $a_{00}$ and a cluster-specific random intercept $u_{0i}$. To fit this hierarchical model, we need to combine the multiple-equation notation into a single-equation format. By substituting (17) into (16) and rearranging the terms, we derive:

$$dq_{it} = a_{00} + u_{0i} + a_1 dX_{it} + a_2 Y_{it} + a_3 dZ_{it} + a_4 Q_o + \varepsilon_{it}$$

$$= a_{00} + a_1 dX_{it} + a_2 Y_{it} + a_3 dZ_{it} + a_4 Q_o + u_{0i} + \varepsilon_{it} \qquad (18)$$

It is important to note that model (18) is equivalent to model (14) with $a_{00}$ corresponding to $a_0$ and $u_{0i}$ corresponding to $u_i$ ($a_{00} \equiv a_0$ and $u_{0i} \equiv u_i$).

(15) is likewise a single-equation specification that incorporates both level-1 and level-2 effects. Similar to (14), its hierarchical representation can be achieved by formulating separate equations for each level, resulting in:

$$dq_{it} = b_{00} + v_{0i} + b_1 dX_{it} + b_2 Y_{it} + b_3 logZ_{it} + b_4 Q_o + \in_{it}$$

$$= b_{00} + b_1 dX_{it} + b_2 Y_{it} + b_3 logZ_{it} + b_4 Q_o + v_{0i} + \in_{it} \qquad (19)$$

where $b_{00} \equiv b_0$ and $v_{0i} \equiv v_i$.

The advantage of incorporating a cluster-specific random intercept into the Bayesian hierarchical regressions is that it enables the estimation of global parameters (shared diffusion and innovation dynamics) while simultaneously capturing heterogeneity at the country level (country-specific intercepts). This approach provides richer insights into how each ASEAN-5 nation deviates from the regional average. Preliminary estimates of development levels suggest that Singapore exhibits a strong innovation coefficient with minimal reliance on catch-up through adoption. Malaysia and Thailand demonstrate a moderate innovation effect and continue to benefit from adoption, though they are gradually shifting towards endogenous innovation. By contrast, Indonesia and the Philippines rely more heavily on technology adoption; their innovation effects are emerging but remain constrained by uneven human capital and infrastructure gaps. For detailed discussions on the structural characteristics of the ASEAN-5 economies, see [90–94].

In Bayesian hierarchical modeling, the selection of prior distributions is crucial for obtaining reliable parameter estimates. For model parameters that can assume both negative and positive values, such as regression coefficients, specifying a mildly informative normal prior distribution, N(0,1), is appropriate. This choice reflects a belief that the parameter values are likely to be near zero but allows for the possibility of larger deviations. For overall variance and variance parameters associated with random effects, it's common to assign prior distributions that reflect non-negativity and incorporate prior beliefs about their scale. A frequently used prior is the inverse-gamma distribution, Ig(0.01, 0.01), which is non-informative and assigns equal plausibility across a wide range of values.

Notably, the high dimensionality of the Bayesian hierarchical model renders the posterior distributions analytically intractable from the prior and likelihood functions. As a result, a hybrid Metropolis-Hastings (MH) sampling algorithm is employed, following the procedure outlined below:

Generate samples $\left\{\theta^{(t)}\right\}_{t=1}^{N}$ from a target distribution $\pi^{(\theta)}$, where $\theta = (\theta_1, \theta_2, \ldots, \theta_d)$.

1. Initialization: Start with an initial value $\theta^{(0)} = \left(\theta_1^{(0)}, \theta_2^{(0)}, \ldots, \theta_d^{(0)}\right)$.

2. Iteration: For each iteration $t$ (from 1 to N):

For each component $i = 1, 2, \ldots, d$ :
Gibbs sampling step (if the conditional is easy to sample from): $\theta_i^{(t)} \sim \pi\left(\theta_i \middle| \theta_{-i}^{(t)}\right)$, where $\theta_{-i}^{(t)} = \left(\theta_1^{(t)}, \theta_{i-1}^{(t)}, \theta_{i+1}^{(t-1)}, \ldots, \theta_d^{(t-1)}\right)$.
MH step (if the conditional is not easy to sample from):
Proposal: Generate a proposed value $\theta_i^*$ from a proposal distribution:
$q(\theta_i^*|\theta_i^{(t-1)})$:

$$\theta_i^* \sim q(\theta_i^*|\theta_i^{(t-1)})$$

Acceptance probability $\alpha$ :

$$\alpha = \min(1, \frac{\pi(\theta_i^*|\theta_{-i}^{(t)})q(\theta_i^{(t-1)}|\theta_i)}{\pi(\theta_i^{(t-1)}|\theta_{-i}^{(t)})q(\theta_i|\theta_i^{(t-1)})})$$

where $\pi(\theta_i^*|\theta_{-i}^{(t)})$ is the target conditional density of $\theta_i$ evaluated at the proposed value $\theta_i^*$, $\pi(\theta_i^{(t-1)}|\theta_{-i}^{(t)})$ is the target conditional density of $\theta_i$ evaluated at the current value $\theta_i^{(t-1)}$, $q(\theta_i^*|\theta_i^{(t-1)})$ is the proposal distribution evaluated at $\theta_i^*$ given the current value $\theta_i^{(t-1)}$, $q(\theta_i^{(t-1)}|\theta_i)$ is the reverse proposal distribution, which is the probability of proposing $\theta_i^{(t-1)}$ from $\theta_i$.

Update: Generate a uniform random number $u \sim U(0, 1)$ and update $\theta_i$:

$$\theta_i^{(t)} = \begin{cases} \theta_i^* & \textit{if } u \leq \alpha \\ \theta_i^{(t-1)} & \textit{if } u > \alpha \end{cases}$$

## Data

Our study employs a panel dataset comprising five advanced ASEAN economies over the period 1965–2019. Data on real GDP, TFP, physical capital, labor, and Mincerian human capital are sourced from the latest Penn World Table (PWT) (version 10.01), updated to 2019. Additional data on human capital measures—such as primary, secondary, and tertiary school enrollment rates, as well as government spending on education—alongside exports as a percentage of GDP and rule of law indicators, are derived from the World Development Indicators (WDI) [72] (Table 1). The export-to-GDP ratio serves as a proxy for trade openness, while the rule of law variable reflects the institutional quality within each country. Owing to data limitations, school enrollment rates are accessible starting from 1971, government education expenditure from 1980, and the rule of law indicator from 1996. Modern growth theories identify trade openness and institutional quality as key proximate determinants of economic growth, complementing traditional factors such as broad capital (both physical and human), labor, and technology [13,14,16,44,95,96]. Among the human capital indicators, primary, secondary, and tertiary school enrollment rates, along with government spending on education, are categorized as flow variables, whereas Mincerian human capital is treated as a stock variable. Notably, Benhabib and Spiegel [6] provided comparable estimates when utilizing both flow and stock measures of human capital. It is noteworthy that this research does not contain any studies with human or animal participants.

## Results and interpretations

**Preliminary analysis.** This subsection first examines the descriptive statistics before transitioning to an analysis of the correlation statistics. Table 2 demonstrates the results of the descriptive statistics, which can be interpreted as follows.

First, growth rates (*dq, dTFP, dK, dL, dH, dP, dS, dT, dE*): *dq* (per capita income growth) and *dTFP* (TFP growth) exhibit notable variation, with mean values of 3.94% and 0.23%, respectively, but wide ranges highlight substantial differences in economic and productivity growth across observations. Growth rates for *dP, dS*, and *dT* (primary, secondary, and tertiary education) and *dH* (Mincerian human capital) are generally moderate, with *dT* showing the highest variability. The variability reflects heterogeneity in educational progress and human capital accumulation. *dE* (government spending on education growth) shows that the wide range and high variability highlight divergent education funding trends across countries, likely influenced by fiscal capacities, political priorities, and responses to economic or demographic pressures.

Second, logarithmic variables (*lnP, lnS, lnT, lnH, lnE*): The logarithmic transformations of education and human capital variables demonstrate relative stability, with low standard deviations. These measures provide a normalized view, indicating that levels of education and human capital are less variable than their growth rates.

Third, initial income ($Q_o$): The mean initial income of 7.85 with a relatively narrow standard deviation suggests that the sample includes countries with somewhat similar starting economic conditions, although the range from 7.26 to 8.64 shows some diversity.

**Table 1. Definition of variables.**

| Variable | | Proxy | Period | Notation | Data source |
|---|---|---|---|---|---|
| Dependent | Growth of per capita income | Annual growth rate of per capita real GDP | 1965–2019 | *dq* | PWT 10.01 |
| Independent | Core | | | | |
| | Growth of Mincerian human capital | Annual growth rate of the index derived from the average years of schooling and the returns to education based on the Mincer equation | 1965–2019 | *dH* | PWT 10.01 |
| | Growth of primary, second-ary, and tertiary education | Annual growth rate of respective schooling enrollment rates | 1971–2019 | *dP, dS, dT* | WDI 2023 |
| | Growth of educational expenditure | Annual growth rate of government spending on education | 1980–2019 | *dE* | WDI 2023 |
| | Logarithm of mean human capital | Natural logarithm of mean human capital indices | 1965–2019 | *lnP, lnS, lnT, lnH, lnE* | PWT. 10.01, WDI 2023 |
| | Initial income per capita | Per capital real GDP | 1964 | $Q_o$ | PWT. 10.01 |
| | Control | | | | |
| | Growth of productivity | Annual growth rate of TFP | 1965–2019 | *dTFP* | PWT 10.01 |
| | Growth of physical capital | Annual growth rate of physical capital stock | 1965–2019 | *dK* | PWT 10.01 |
| | Growth of labor | Annual growth rate of people engaged in an economy | 1965–2019 | *dL* | PWT 10.01 |
| | Trade openness | Export as a percentage of GDP | 1965–2019 | *Ex* | WDI 2023 |
| | Institutional quality | Rule of law | 1996–2019 | *Ru* | WDI 2023 |

*Source: Compilation and calculation by the author.*

**Table 2. Descriptive statistics.**

| Variable | Obs | Mean | Std. dev. | Min | Max |
|---|---|---|---|---|---|
| *dq* | 275 | 3.936578 | 4.105242 | −14.34718 | 34.17879 |
| *dTFP* | 275 | 0.2290056 | 3.679473 | −16.98565 | 30.96437 |
| *dK* | 275 | 6.840325 | 3.359937 | −0.6917631 | 17.17097 |
| *dL* | 275 | 2.759731 | 2.371208 | −3.055798 | 12.44507 |
| *dH* | 275 | 1.355774 | 0.9744951 | −0.6057407 | 4.744653 |
| *dP* | 182 | 0.1181616 | 2.152985 | −4.933402 | 12.78346 |
| *dS* | 182 | 2.377262 | 5.455952 | −12.74672 | 51.54628 |
| *dT* | 161 | 5.556977 | 14.80183 | −26.1229 | 143.9035 |
| *dE* | 137 | 0.914258 | 12.27027 | −25.21008 | 50.91954 |
| *lnH* | 275 | 0.7381407 | 0.076477 | 0.6171248 | 0.820696 |
| *lnP* | 275 | 4.636435 | 0.0352584 | 4.598537 | 4.686046 |
| *lnS* | 275 | 4.246672 | 0.2440265 | 3.975843 | 4.681448 |
| *lnT* | 275 | 3.380523 | 0.6035083 | 2.631654 | 4.467944 |
| *lnE* | 275 | 1.201996 | 0.2647791 | 0.960396 | 1.692376 |
| $Q_o$ | 275 | 7.853624 | 0.5047535 | 7.256256 | 8.63734 |
| *Ex* | 259 | 70.08194 | 58.88137 | 5.276255 | 228.9938 |
| *Ru* | 105 | 0.2026924 | 0.8110808 | −0.9102647 | 1.837814 |

*Source: Calculation by the author.*

Fourth, export share (*Ex*): The export share exhibits substantial variability, with a wide range from 5.28% to 229%. This reflects significant differences in the economic structures of the countries, with some heavily reliant on exports.

Fifth, rule of law (*Ru*): The rule of law variable has a mean close to zero, indicating a balance between positive and negative governance environments across the sample. Its wide range suggests notable diversity in institutional quality.

Overall, the data highlights significant heterogeneity in economic growth, productivity, education, and institutional variables, offering a comprehensive foundation for analyzing their impacts on economic performance.

Table 3 highlights significant multicollinearity among the variables in our models, a challenge that conventional frequentist techniques struggle to address, often leading to estimation biases. Specifically, initial income ($Q_o$) is strongly associated with the three human capital measures (*lnH, lnS, lnT*), with respective correlation coefficients of 0.9, 0.96, and 0.8.

**Table 3. Correlation statistics.**

| | dq | dTFP | dK | dL | dH | dP | dS |
|---|---|---|---|---|---|---|---|
| dq | 1.0000 | | | | | | |
| dTFP | 0.7565 | 1.0000 | | | | | |
| dK | 0.5073 | −0.0046 | 1.0000 | | | | |
| dL | 0.2239 | −0.1743 | 0.2415 | 1.0000 | | | |
| dH | −0.0495 | −0.2076 | 0.1097 | 0.0927 | 1.0000 | | |
| dP | −0.0288 | −0.0788 | −0.0159 | 0.0394 | 0.0464 | 1.0000 | |
| dS | −0.0287 | −0.0575 | 0.1630 | −0.0521 | −0.0992 | 0.2169 | 1.0000 |
| dT | 0.0260 | 0.0455 | 0.1632 | 0.0641 | 0.1555 | 0.2132 | 0.0339 |
| dE | −0.2917 | −0.2199 | 0.0429 | −0.1416 | 0.0259 | 0.0138 | 0.0504 |
| lnH | −0.0130 | −0.1179 | 0.0994 | 0.1391 | 0.1683 | −0.0915 | −0.1983 |
| lnP | −0.1795 | −0.0615 | −0.2145 | 0.0191 | −0.1769 | −0.0510 | −0.0242 |
| lnS | 0.0716 | −0.0934 | 0.1647 | 0.1815 | 0.2598 | −0.0914 | −0.2118 |
| lnT | 0.1249 | −0.0139 | 0.1769 | 0.1256 | 0.2690 | −0.0352 | −0.0833 |
| lnE | 0.0787 | −0.0336 | 0.1542 | 0.0001 | 0.0736 | 0.0192 | −0.0640 |
| $Q_o$ | 0.0119 | −0.1387 | 0.1226 | 0.1962 | 0.2145 | −0.1113 | −0.2448 |
| Ex | 0.0659 | −0.0706 | 0.0668 | 0.1738 | 0.3141 | −0.0595 | −0.1602 |
| Ru | −0.0260 | −0.1504 | 0.0358 | 0.2535 | 0.6490 | 0.1130 | −0.1494 |
| | dT | dE | lnH | lnP | lnS | lnT | lnE |
| dT | 1.0000 | | | | | | |
| dE | 0.0986 | 1.0000 | | | | | |
| lnH | −0.1193 | 0.0020 | 1.0000 | | | | |
| lnP | −0.0965 | 0.0178 | −0.3565 | 1.0000 | | | |
| lnS | −0.1231 | −0.0086 | 0.8608 | −0.2627 | 1.0000 | | |
| lnT | −0.0144 | −0.0288 | 0.7148 | −0.4283 | 0.9177 | 1.0000 | |
| lnE | 0.0344 | 0.0213 | 0.3053 | −0.7450 | 0.0151 | −0.0591 | 1.0000 |
| $Q_o$ | -0.1618 | 0.0040 | 0.9018 | −0.1075 | 0.9603 | 0.7752 | 0.0417 |
| Ex | −0.0844 | −0.0188 | 0.6547 | −0.3552 | 0.8742 | 0.8459 | 0.0687 |
| Ru | 0.0187 | −0.0892 | 0.6512 | −0.5444 | 0.8629 | 0.8762 | 0.2271 |
| | $Q_o$ | Ex | Ru | | | | |
| $Q_o$ | 1.0000 | | | | | | |
| Ex | 0.7912 | 1.0000 | | | | | |
| Ru | 0.7427 | 0.9380 | 1.0000 | | | | |

*Source: Calculation by the author.*

Moreover, $Q_o$ is highly correlated to the additional controls *Ex* (0.8) and *Ru* (0.7). The additional controls (*Ex, Ru*) themselves show substantial interdependence, with a correlation coefficient of 0.9, and they are highly related to the human capital indices. Specifically, the correlation coefficients between *Ex* and the human capital measures (*lnS* and *lnT*) are 0.9 and 0.8, respectively, while those between *Ru* and the same measures are 0.9. In this context, the Bayesian approach offers an efficient and flexible framework for addressing multicollinearity, leveraging informative priors and robust estimation techniques to mitigate the biases inherent in traditional methods.

**Main analysis.** First and foremost, we develop a traditional growth accounting framework in which human capital is modeled as an independent variable (see (14)). In this specification, the growth of per capita real income serves as the dependent variable, while the independent variables include the growth of TFP, physical capital, and human capital, alongside initial income. Various model specifications are estimated using different human capital measurements—namely, primary, secondary, and tertiary schooling, Mincerian human capital, and government spending on education. The analysis proceeds in stages: initially, we examine the correlation between human capital growth and per capita income growth; subsequently, the initial income variable($Q_o$) is incorporated; and finally, both primary control variables (growth of TFP, physical capital, and labor) and additional controls (trade openness and institutional quality) are included. The findings reported in Table 4 indicate that while the effects of control variables remain consistent across various human capital indices, the direction of the relationship between human capital growth and per capita income growth varies. Furthermore, except for government spending on education, the initial income variable ($Q_o$) demonstrates a negative coefficient across the other human capital indicators. Specifically, the results for the human capital variables are mixed: primary schooling and the Mincerian indicator show positive but ambiguous effects (with estimated probabilities below 0.7), while secondary schooling, tertiary schooling, and government spending on education exhibit negative effects. This outcome is consistent with Benhabib and Spiegel [6], who also found that alternative human capital measures often yield negative or insignificant effects. These findings imply that the traditional growth accounting framework is misspecified and inadequate for capturing the substantial role of human capital in long-term economic growth—a point emphasized theoretically by [12] and supported empirically by Verspagen [97], Benhabib and Spiegel [6], Howitt and Mayer-Foulkes [98], AtiqurRahman and Zaman [41], and Knez [7]. S1-5 Tables in Supporting information provide a more comprehensive presentation of the estimation results.

Furthermore, using (15), we estimate the extended Nelson-Phelps model through a three-step process. Initially, we examine the correlation between per capita income, as the dependent variable, and the log of mean human capital indices alongside the primary control variables. In the second step, the initial income variable($Q_o$)is incorporated, and finally, all control variables are included in the analysis. The results, summarized in Table 5, reveal that while the positive elasticities of the primary controls and the negative slope coefficients of the human capital variables remain consistent initially, these correlations exhibit instability upon the inclusion of the initial income variable ($Q_o$). However, when all control variables are integrated, the results stabilize, except for the primary schooling (*lnP*) and government spending on education (*lnE*) variables. Notably, the slope coefficients for the logarithm of mean human capital indices assume the expected positive sign, whereas the initial income variable consistently displays a negative slope coefficient, aligning with theoretical expectations. Regarding the additional controls, the trade openness (*Ex*) variable exhibits a negative slope coefficient, while the

**Table 4. Traditional growth accounting.**

| Specification | dP | dS | dT | dH | dE | $Q_o$ |
|---|---|---|---|---|---|---|
| Additional controls excluded | + | − | − | + | − | |
| $Q_o$ included | + | − | − | + | − | − |
| All controls included | + | − | − | + | − | -* |

*Note: * denotes that the slope coefficient of "initial income (<<Eqn136>>)" takes a positive value for "government expenditure on education".*

*Source: Calculation by the author.*

**Table 5. Estimating the extended Nelson-Phelps model.**

| Specification | lnP | lnS | lnT | lnH | lnE | $Q_o$ | Ex | Ru |
|---|---|---|---|---|---|---|---|---|
| Additional controls excluded | – | – | – | – | – | | | |
| $Q_o$ included | – | + | + | – | + | -* | | |
| All controls included | + | + | + | + | + | -** | – | + |

Note: * and ** notice that the coefficient for "initial income" takes a positive value for "primary school" and "government expenditure on education", respectively.

Source: Calculation by the author.

institutional factor (*Ru*) demonstrates a positive coefficient. Comprehensive results are detailed in S6-10 Tables in Supporting information.

From the above empirical analyses, we derive intermediate conclusions as follows. The results (Tables 5, 6) reveal consistent effects across all human capital measures. The growth of fundamental factors such as TFP, physical capital, and labor shows strong positive effects on productivity growth with the highest probability (1). For human capital, the positive effects of schooling at secondary and tertiary levels are most pronounced, with respective probabilities of 0.71 and 0.88, while primary education shows a weaker positive effect (0.57). In contrast, the effects of Mincerian human capital and government spending on education are ambiguous, with probabilities of 0.51 and 0.54, respectively. Initial income($Q_o$) exhibits strongly negative effects for secondary and tertiary education levels (probabilities of 0.70 and 0.81) but remains ambiguously negative for Mincerian human capital and government spending on education.

Additionally, the probabilities of all human capital measures' effects are higher for log mean human capital than for initial income, indicating that ASEAN-5 economies have transitioned from technology adoption to national innovation. Among the additional controls, institutional factors (*Ru*) show consistently positive effects across all human capital indices (probabilities > 0.7), while trade openness (*Ex*) has a surprising negative effect. This may be attributed to the export-led growth strategy in ASEAN economies, where low- and medium-technology exports dominate while imports drive exports. This reliance may delay the shift toward high-technology capital goods exports [99].

**Diagnostics of MCMC convergence.** When employing a MCMC sampler, it is essential to assess chain convergence to ensure that model parameters have stabilized at plausible values. A trace plot serves as an effective visual tool for this purpose, depicting the sampled values of a parameter across iterations in the MCMC simulation. By illustrating how a parameter's value evolves throughout the simulation, trace plots help determine whether the MCMC chain has stabilized and is sampling from the target posterior distribution. Mathematically, a trace plot is a function:

**Table 6. Probability of effect.**

| Coefficient | Primary education | Secondary education | Tertiary education | Mincerian human capital | Government spending on education |
|---|---|---|---|---|---|
| dTFP (positive) | 1 | 1 | 1 | 1 | 1 |
| dK (positive) | 1 | 1 | 1 | 1 | 1 |
| dL (positive) | 1 | 1 | 1 | 1 | 1 |
| Logarithm of mean human capital index (positive) | 0.5715 | 0.7126 | 0.8759 | 0.5114 | 0.5448 |
| $Q_o$ (negative) | 0.5691 | 0.6952 | 0.808 | 0.4987 | 0.4863 |
| Ex (negative) | 0.8274 | 0.8592 | 0.925 | 0.8557 | 0.8617 |
| Ru (positive) | 0.8458 | 0.827 | 0.687 | 0.8094 | 0.7895 |
| intercept (positive) | 0.5267 | 0.5347 | 0.5317 | 0.5581 | 0.5311 |

Source: Calculation by the author.

$$f(i) = \theta^{(i)} \tag{20}$$

where $i$ denote the iteration index ($i = 1, 2, 3, \ldots, N$), $N$ represents the total number of iterations, and $\theta^{(i)}$ denote the sampled value of the parameter $\theta$ at iteration $i$. Graphically, each point on the trace plot corresponds to the coordinates ($i$, $\theta^{(i)}$), with the iteration index $i$ plotted on the x-axis and the sampled value $\theta^{(i)}$ on the y-axis.

We assess chain convergence for all model parameters; however, for illustrative purposes, we present the trace plots for the key variables across the human capital measurements: the logarithm of mean human capital and initial income. In Fig 3, the trace plots exhibit oscillations around a stable mean without any discernible trends, indicating that the chains have converged to the target posterior distribution. This pattern suggests that the MCMC process has effectively captured the posterior distribution, providing strong evidence of reliability and interpretability in the Bayesian results.

## Conclusion

This study investigates the dual role of human capital within an extended Nelson-Phelps technology catch-up framework, enriched by Romer's Schumpeterian growth model, using a panel dataset of five advanced ASEAN economies (ASEAN-5) from 1965 to 2019. It makes some key contributions to growth literature.

First, from a methodological perspective, this study addresses statistical challenges such as multicollinearity and reverse causation through the adoption of a Bayesian hierarchical approach. This method incorporates specific informative priors for structural parameters in regression models, enabling it to capture the interactions, impact direction, and magnitude of predictor variables on the dependent variable. These features help mitigate potential statistical complexities such as multicollinearity and reverse causation [85–88]. A Bayesian hierarchical approach can also produce reliable estimates through shrinkage effects [71]. Additionally, by incorporating productivity (TFP) as an independent variable, this study overcomes omitted variable bias, a limitation in similar studies [6,32,41]. Using a panel framework effectively addresses heterogeneity issues inherent in Barro-type cross-country growth regressions, which often lead to biased estimates [68–70,100]. Furthermore, the Bayesian hierarchical approach not only circumvents the reliance on p-values—now

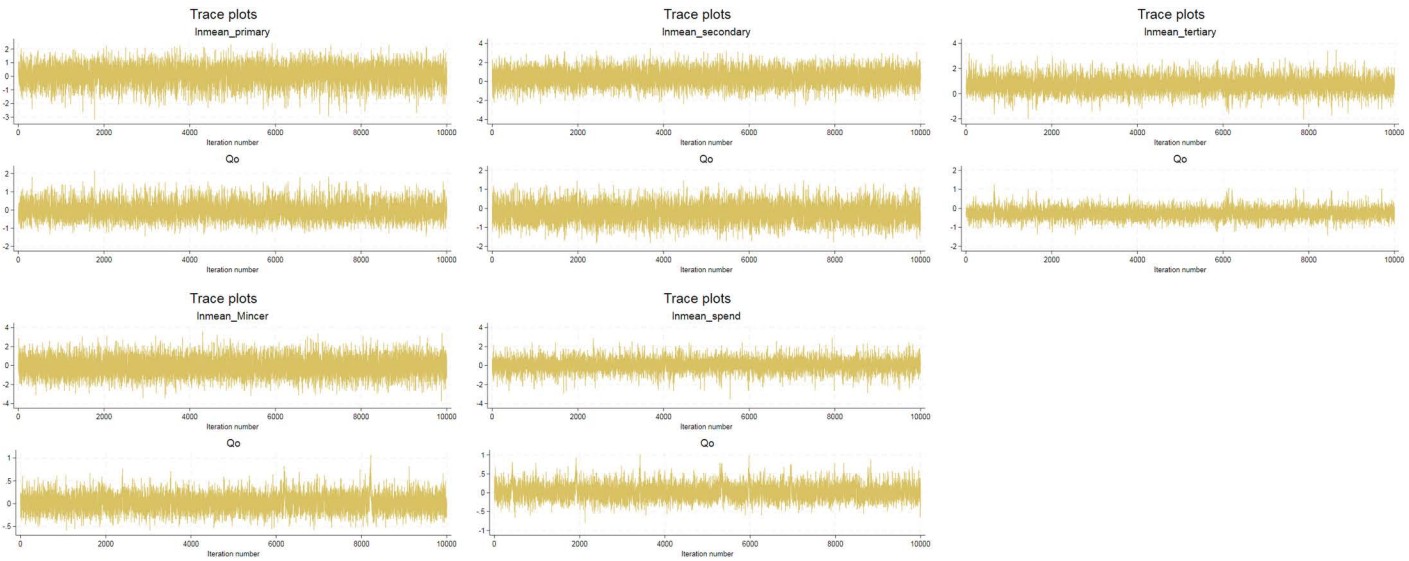

**Fig 3. Trace Plots for Assessing MCMC Convergence.** *Source: Calculation by the author.*

considered debatable due to their potential to produce misleading conclusions [74–76], but also takes into account country-specific effects [89].

Second, on the theoretical front, this study reaffirms the limitations of the traditional accounting model, where human capital is treated as an independent predictor. As documented by Nelson and Phelps [12] and empirically validated by Benhabib and Spiegel [6], this specification is a misrepresentation of the role of human capital in the growth process. By contrast, the study validates the dual role of human capital, as theorized in the Nelson-Phelps and Romer models, in enhancing productivity in ASEAN-5 economies. The study identifies the dual contribution of aggregate human capital in fostering domestic innovation and facilitating the adoption and absorption of foreign technologies, with the former becoming increasingly significant. Additionally, by conducting interval hypothesis tests, the study distinguishes the role of different education levels in generating national innovation and adopting foreign technologies. The results demonstrate that skilled workers with secondary and tertiary education contribute significantly more to innovation and technology catch-up than low-skilled workers.

Finally, regarding policy implications, the extended Nelson-Phelps model of technology catch-up emerges as an appropriate growth framework for ASEAN economies. This model supports their transition from low- and medium-technology-based advantages to diversified, high-technology-export economies, where domestic innovation takes precedence over foreign technology imports. Evidence suggests an improvement in the technological content of ASEAN manufactured exports, reflecting a catch-up process driven by technology transfer and innovation. By employing a robust Bayesian estimation method, this study provides a solid empirical foundation for designing effective growth policies not only within the ASEAN community but also for other emerging and developing economies.

As ASEAN economies transition from adoption-led to innovation-led growth, human capital strategies must shift from providing basic absorptive capacity to cultivating frontier innovation skills. A phased approach—first strengthening foundational skills for adoption, then upgrading to advanced R&D and innovation capabilities—ensures sustained convergence toward the technological frontier.

For economies where national innovation has become the dominant mechanism of technological advancement, human capital policies should emphasize advanced skill formation, research capability, and the integration of education with industrial innovation systems. Governments should expand graduate and doctoral programs in frontier fields such as artificial intelligence, biotechnology, and advanced materials, often through partnerships with leading global research universities that can co-develop curricula and joint PhD programs. Beyond technical specialization, innovation also depends on fostering entrepreneurial and cross-disciplinary skills, which can be achieved by embedding innovation-focused entrepreneurship education and design thinking into universities and technical institutes, as well as supporting student and faculty startups with seed funding and incubator programs. In parallel, private-sector R&D must be stimulated through tax incentives, matching grants, and joint research centers that connect industrial clusters—such as electronics in Malaysia or agritech in Thailand—with local universities, while continuous professional development should be provided to build expertise in innovation management and intellectual property.

For economies where technology adoption from abroad remains central, particularly those ASEAN members transitioning toward innovation-led growth, human capital strategies must focus on building absorptive capacity and applied skills necessary to integrate external technologies effectively. This begins with strengthening foundational and technical education, including universal secondary schooling and vocational training aligned with the requirements of imported technologies, such as automation and digital manufacturing. It also requires improving digital literacy and managerial capabilities so that the workforce can not only operate advanced tools but also adapt organizational processes for efficient technology integration; this includes training in areas like lean manufacturing and global value chain participation. Finally, technology transfer should be facilitated through practical mechanisms: apprenticeship and on-the-job learning within foreign-invested firms, requirements for technology-sharing components in foreign direct investment agreements, and public–private partnerships that co-develop training programs tied to multinational operations.

 

In addition, accounting for structural differences [90–94], we propose specific policy recommendations tailored to each ASEAN-5 country. Singapore, as a high-income innovation hub with advanced R&D infrastructure and leading universities, derives its growth primarily from frontier innovation rather than imitation. Policy should therefore focus on frontier research, particularly in areas such as artificial intelligence and biotechnology, while sustaining policies that attract global talent. Malaysia's strong manufacturing base and electronics and electrical (E&E) sector are offset by uneven innovation ecosystems outside the Klang Valley. Accordingly, policy should strengthen university–industry linkages and prioritize upgrading supply chains toward innovation-driven production. In Thailand, innovation is playing an increasingly important role, though technology adoption remains significant; its strengths in automotive and agritech are supported by moderate R&D intensity, which is catching up to Malaysia and Singapore. Policy should promote cluster-based R&D, especially in the Eastern Economic Corridor, and invest in STEM workforce upskilling. Indonesia benefits from a large domestic market and abundant natural resources but faces constraints due to fragmented R&D institutions and low tertiary enrollment in STEM. Policy priorities should include investing in foundational education and developing regional innovation hubs, particularly beyond Java. Finally, the Philippines leverages strengths in service exports and English proficiency but suffers from a weak industrial R&D base and limited patent activity. Policy should therefore foster innovation in services—such as fintech and healthtech—and mitigate brain drain through incentives that encourage return migration.

The study's primary limitation is its focus on five advanced ASEAN countries, excluding other member nations due to data unavailability. This narrow scope may limit the generalizability of the findings. Expanding the research to include all emerging and developing economies would provide a more comprehensive understanding of human capital's role in technology adoption and innovation. Such an expansion is essential from both scholarly and practical perspectives, as it would offer insights applicable to a broader range of economic contexts.

## Supporting information

**S1 Table. Traditional growth accounting (Primary school).**
(PDF)

**S2 Table. Traditional growth accounting (Secondary school).**
(PDF)

**S3 Table. Traditional growth accounting (Tertiary school).**
(PDF)

**S4 Table. Traditional growth accounting (Government spending on education).**
(PDF)

**S5 Table. Traditional growth accounting (Mincerian human capital).**
(PDF)

**S6 Table. Estimating extended Nelson-Phelps model (Primary school).**
(PDF)

**S7 Table. Estimating extended Nelson-Phelps model (Secondary school).**
(PDF)

**S8 Table. Estimating extended Nelson-Phelps model (Tertiary school).**
(PDF)

**S9 Table. Estimating extended Nelson-Phelps model (Mincerian human capital).**
(PDF)

**S10 Table. Estimating extended Nelson-Phelps model (Government spending on education).**
(PDF)

## Acknowledgments

The authors would like to thank the anonymous reviewers and the editors for their valuable comments and suggestions.

## Author contributions

**Conceptualization:** Nguyen Ngoc Thach.

**Data curation:** Nguyen Ngoc Thach.

**Formal analysis:** Nguyen Ngoc Thach.

**Investigation:** Nguyen Ngoc Thach.

**Methodology:** Nguyen Ngoc Thach.

**Project administration:** Nguyen Ngoc Thach.

**Resources:** Nguyen Ngoc Thach.

**Software:** Nguyen Ngoc Thach.

**Supervision:** Nguyen Ngoc Thach.

**Validation:** Nguyen Ngoc Thach.

**Visualization:** Nguyen Ngoc Thach.

**Writing – original draft:** Nguyen Ngoc Thach.

**Writing – review & editing:** Nguyen Ngoc Thach.

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
