## [Decision Letter · Decision Letter 0]

1 Aug 2025

Dear Thach Nguyen Ngoc,

Thank you for submitting your manuscript to PLOS ONE. After careful consideration, we feel that it has merit but does not fully meet PLOS ONE’s publication criteria as it currently stands. Therefore, we invite you to submit a revised version of the manuscript that addresses the points raised during the review process.

https://journals.plos.org/plosone/s/submission-guidelines#loc-laboratory-protocols . Additionally, PLOS ONE offers an option for publishing peer-reviewed Lab Protocol articles, which describe protocols hosted on protocols.io. Read more information on sharing protocols at https://plos.org/protocols?utm_medium=editorial-email&utm_source=authorletters&utm_campaign=protocols .

We look forward to receiving your revised manuscript.

Kind regards,

Rogis Baker, Ph.D

Academic Editor

PLOS ONE

Journal Requirements:

“The authors have no competing interests.”

4. Please note that your Data Availability Statement is currently missing the DOI/accession number of each dataset or a direct link to access each database. If your manuscript is accepted for publication, you will be asked to provide these details on a very short timeline. We therefore suggest that you provide this information now, though we will not hold up the peer review process if you are unable.

Reviewers' comments:

Reviewer's Responses to Questions

**Comments to the Author**

1. Is the manuscript technically sound, and do the data support the conclusions?

Reviewer #1: Yes

Reviewer #2: Yes

2. Has the statistical analysis been performed appropriately and rigorously?

Reviewer #1: Yes

Reviewer #2: Yes

3. Have the authors made all data underlying the findings in their manuscript fully available?

Reviewer #1: Yes

Reviewer #2: Yes

4. Is the manuscript presented in an intelligible fashion and written in standard English?

Reviewer #1: Yes

Reviewer #2: Yes

Reviewer #1: The paper presents a valuable contribution to the literature on technology diffusion and economic growth in ASEAN economies. However, to enhance its theoretical coherence, empirical rigor, and policy relevance, several areas require further refinement and clarification:

1. In 3.1 section, while the paper claims to integrate Romer’s Schumpeterian model into the Nelson-Phelps framework, it would benefit from a clearer and more explicit theoretical exposition of how the two are operationalized jointly in the model. A conceptual diagram could be helpful here.

2. Given the structural differences among the ASEAN-5 economies, more discussion on country-specific effects or sub-sample analyses would enrich the interpretation of the results and highlight policy implications tailored to individual countries.

3. In 4.2 section , the paragraph reports that human capital growth has opposing signs depending on the measure used, positive for primary schooling and Mincerian indicators, but negative for secondary, tertiary, and government spending on education, without offering a clear economic rationale.

4. In the conclusion part, The ASEAN region is highly heterogeneous , yet the paragraph treats it as a uniform entity.

Reviewer #2: Adopting a Bayesian hierarchical approach, this manuscript investigates the dual role of human capital---fostering national innovation and facilitating the adoption of external technologies ---within an extended Nelson-Phelps technology catch-up framework, enriched by Romer's Schumpeterian growth model, using a panel dataset of five advanced ASEAN economies (ASEAN-5) from 1965 to 2019.

The manuscript is technically sound, and the data supports the conclusions. The statistical analysis has been performed appropriately and rigorously. The author declared that all data are fully available without restriction. The manuscript is presented in an intelligible fashion and written in standard English.

But there are some parts of the manuscript need to be modified:

1.The subsection of “Related empirical studies” presented the literature is somewhat dated , please supplement the latest literature.

2.In the model 14 and model 15, control variables have been divided into primary control variables and additional control variables. What is the meaning of doing so? Can the two control variables be combined? If they cannot be combined, please explain the reasons in the text.

3.The core explanatory variables and control variables are not distinguished in the table 1 (Definition of variables).

4.The proposal for developing human capital is not based on the results of the study and is disjointed from them. The author should make targeted recommendations closely related to the research results.

**Do you want your identity to be public for this peer review?** For information about this choice, including consent withdrawal, please see our Privacy Policy

Reviewer #1: No

Reviewer #2: No

---

## [Author Response · Author response to Decision Letter 1]

5 Sep 2025

Dear Respected Reviewers and Editors,

Thank you very much for your valuable suggestions and comments. In response, we have thoroughly revised our manuscript. Please let us know if you have any further questions or additional feedback.

Reviewer #1: The paper presents a valuable contribution to the literature on technology diffusion and economic growth in ASEAN economies. However, to enhance its theoretical coherence, empirical rigor, and policy relevance, several areas require further refinement and clarification:

1. In 3.1 section, while the paper claims to integrate Romer’s Schumpeterian model into the Nelson-Phelps framework, it would benefit from a clearer and more explicit theoretical exposition of how the two are operationalized jointly in the model. A conceptual diagram could be helpful here.

Response:

We have clarified this content in Section “Analytical framework” and for clarity, added a conceptual diagram in Fig 2 (please see green highlights on p. 6).

2. Given the structural differences among the ASEAN-5 economies, more discussion on country-specific effects or sub-sample analyses would enrich the interpretation of the results and highlight policy implications tailored to individual countries.

Response:

We have discussed more about country-specific effects in the Methodology and Conclusion sections (please see green highlights on pp. 13-14).

3. In 4.2 section, the paragraph reports that human capital growth has opposing signs depending on the measure used, positive for primary schooling and Mincerian indicators, but negative for secondary, tertiary, and government spending on education, without offering a clear economic rationale.

Response:

This finding arises from the traditional growth accounting framework, which has been shown to be misspecified (see green highlights on p. 19). We therefore rely on the estimates derived from the Nelson–Phelps catch-up model as our main results, which are reported in the second point of the conclusion (see green highlights on p. 22).

4. In the conclusion part, The ASEAN region is highly heterogeneous, yet the paragraph treats it as a uniform entity.

Response:

The conclusion of the study highlights the dual role of aggregate human capital in both fostering domestic innovation and facilitating the adoption and absorption of foreign technologies, with the former becoming increasingly important in the ASEAN context. Accordingly, two sets of policy measures are proposed: (i) common measures applicable to the ASEAN community (see yellow highlights on pp. 23–24) and (ii) country-specific measures tailored to each of the ASEAN-5 nations (see green highlights on p. 24).

Reviewer #2: Adopting a Bayesian hierarchical approach, this manuscript investigates the dual role of human capital---fostering national innovation and facilitating the adoption of external technologies ---within an extended Nelson-Phelps technology catch-up framework, enriched by Romer's Schumpeterian growth model, using a panel dataset of five advanced ASEAN economies (ASEAN-5) from 1965 to 2019.

The manuscript is technically sound, and the data supports the conclusions. The statistical analysis has been performed appropriately and rigorously. The author declared that all data are fully available without restriction. The manuscript is presented in an intelligible fashion and written in standard English.

But there are some parts of the manuscript need to be modified:

1. The subsection of “Related empirical studies” presented the literature is somewhat dated, please supplement the latest literature.

Response:

We added the latest studies (please see green studies cited in the text).

2. In the model 14 and model 15, control variables have been divided into primary control variables and additional control variables. What is the meaning of doing so? Can the two control variables be combined? If they cannot be combined, please explain the reasons in the text.

Response:

Traditionally in growth literature, technology, broad capital, and labor have been regarded as the fundamental drivers of economic growth (e.g., Solow, 1956; Mankiw et al., 1992). Consequently, in diffusion models where the level or growth of human capital serves as the core independent variable, these factors are typically included as primary covariates (e.g., Benhabib & Spiegel, 1964), while variables such as institutions, openness, or geographical dummies are treated as supplementary controls. This distinction stems from an economic perspective but does not materially affect the econometric outcomes. However, to avoid unnecessary complexity for readers and to present our model more clearly, we include all of these variables as control variables (see blue highlights on pp. 13–14).

3. The core explanatory variables and control variables are not distinguished in the table 1 (Definition of variables).

Response:

We have revised Table 1 (p. 16).

4. The proposal for developing human capital is not based on the results of the study and is disjointed from them. The author should make targeted recommendations closely related to the research results.

Response:

The study identifies the dual contribution of aggregate human capital in fostering domestic innovation and facilitating the adoption and absorption of foreign technologies, with the former becoming increasingly significant in the ASEAN context. So, the study offers two sets of measurements: common and specific.

i) Common measurements for the ASEAN region (please see yellow highlights on p. 23-24)

ii) Specific to each ASEAN-5 country (please see green highlights on p. 24)

---

## [Editor Report · Decision Letter 1]

19 Sep 2025

Human Capital's Dual Impact: Advancing Innovation and Technology Diffusion in ASEAN-5 through the Nelson-Phelps-Romer Lens

PONE-D-25-07521R1

Dear Ngoc,

We’re pleased to inform you that your manuscript has been judged scientifically suitable for publication and will be formally accepted for publication once it meets all outstanding technical requirements.

Kind regards,

Rogis Baker, Ph.D

Academic Editor

PLOS ONE
---

## [Editor Report · Acceptance letter]

PONE-D-25-07521R1

PLOS ONE

Dear Dr. Ngoc Thach,

I'm pleased to inform you that your manuscript has been deemed suitable for publication in PLOS ONE. Congratulations! Your manuscript is now being handed over to our production team.

Kind regards,

on behalf of

Dr. Rogis Baker

Academic Editor

PLOS ONE